# BINDING VISUAL FEATURES POINT BY POINT

## ABSTRACT

Despite success on standard benchmarks, vision language models display persistent failures on tasks involving processing of multi-object scenes, including many tasks that are relatively easy for humans. Recent work has found that these failures may stem from a basic inability to accurately bind object features in-context, a challenge that is referred to as the 'binding problem' in cognitive science and neuroscience. The human visual system is thought to solve this binding problem via serial processing, attending to individual objects one at a time so as to avoid interference from other objects. Here, we investigate 'pointing' – the use of explicit spatial coordinates to refer to objects – as an analogous solution for vision language models. We find that learning to point-via-text induces an internal visual search routine, and we characterize the mechanisms that support this procedure. We also find that pointing behavior can be generalized to new tasks via fine-tuning, and that doing so eliminates binding errors and enables compositional generalization. These results provide a proof-of-principle that serial processing can solve the binding problem for vision language models just as it does for biological vision.

## 1 INTRODUCTION

Vision Language Models (VLMs) have achieved impressive performance on a range of multimodal tasks (OpenAI & et al., 2024a; Ramesh et al., 2022), yet they continue to struggle with tasks involving multi-object scenes, including counting (Rahmanzadehgervi et al., 2024; Rane et al., 2024; Zhang & Wang, 2024), relational image generation (Conwell & Ullman, 2022), relational scene understanding (Lewis et al., 2022; Thrush et al., 2022; Fu et al., 2025), and visual analogy (Mitchell et al., 2023; Yiu et al., 2024), even in settings with near-perfect human accuracy. Recent work has identified the *binding problem* as a key factor underlying many of these observed failure modes (Campbell et al., 2024). The binding problem arises in any setting where a shared set of features must be associated (in-context) with a set of distinct entities (Greff et al., 2020). For instance, in a scene with multiple colored shapes, binding is needed to precisely represent the associations between shape and color, and prevent *binding errors* (e.g., erroneously identifying a blue circle in an image that contains a blue square and a red circle). The persistent failure of VLMS on tasks that require in-context binding suggests that addressing the binding problem is an important priority for improving VLM performance.

There is a long history of research on the binding problem in cognitive science and neuroscience (Roskies, 1999; Treisman & Gelade, 1980; Von Der Malsburg, 1994; Frankland et al., 2021b), and this literature may offer useful clues for addressing the binding problem in VLMs. A central finding of this literature is that human visual processing is characterized by two distinct modes. When forced to process images rapidly, human vision is largely limited to a feedforward, parallel mode, in which visual judgments are highly susceptible to interference between similar objects, often resulting in binding errors (Treisman & Gelade, 1980; Treisman & Schmidt, 1982). However, the human visual system can overcome these limitations via *serial processing* (Treisman & Gelade, 1980; Roelfsema, 2023). For instance, in a visual search task involving objects with shared features (e.g., a small set of shapes and colors), human observers can avoid interference by directing attention to individual objects one at a time. This raises the question of whether the binding problem in VLMs may be resolved via a similar form of serial processing.

How might VLMs be augmented with a capacity for serial visual processing, akin to the sequential visual attention that enables binding in human vision? One possibility is to employ the 'pointing' procedure recently introduced by the Molmo family of open-source VLMs (Deitke et al., 2024). In

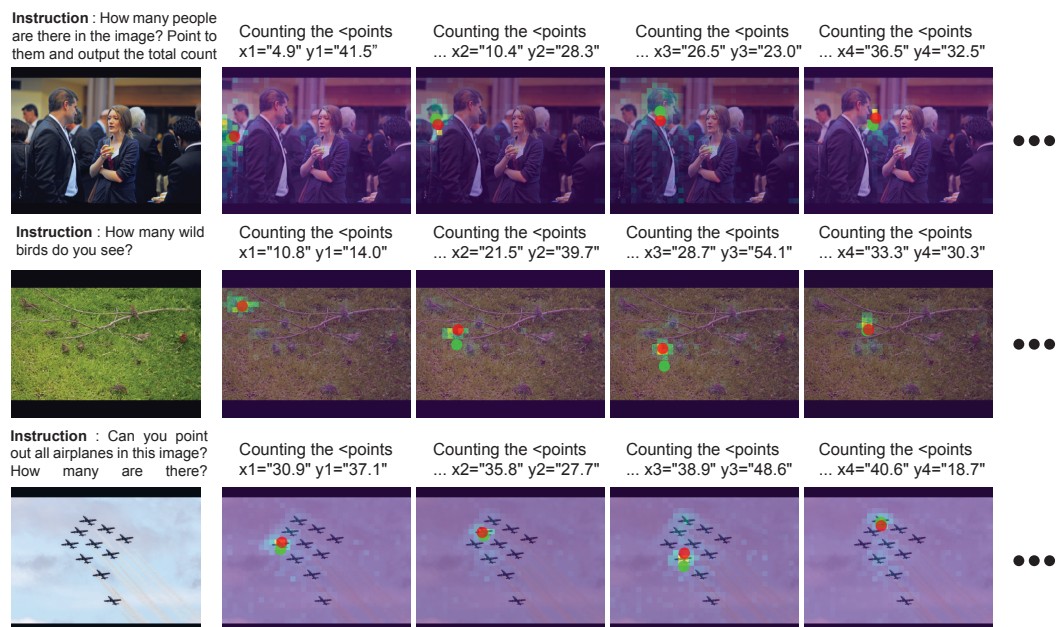

Figure 1: **Pointing-via-text induces an internal visual search routine in VLMs.** The left-most column shows example images and corresponding prompts. Columns 2–5 illustrate the pointing trace generated by the Molmo vision language model, along with the model's internal attentional states. Each frame displays the average distribution of attention accompanying the generation of coordinates for a single object. The red dots indicate the centroids of these attention distributions, and the green dots indicate the coordinates generated in the model's textual output. The results show a close match between textual coordinates and internal attentional states, despite the lack of explicit supervision on these internal states.

that approach, VLMs are trained to 'point' to objects one at a time by providing explicit spatial coordinates (i.e., positions along the x and y axes) for each object. For instance, when asked to count the objects in an image, a pointing trace would enumerate the spatial coordinates of each object before providing a final count (Figure 1). Although this procedure has obvious similarities to human serial visual attention, it is unclear whether learning to point actually induces an internal visual search routine, or whether this procedure can resolve the binding problem for VLMs.

In this work, we present a comprehensive investigation of pointing-via-text as a candidate solution to the binding problem in VLMs. We first investigate the mechanistic and representational origins of binding errors in VLMs, finding that these are linked to attentional interference between objects with shared features. We then conduct a mechanistic analysis of Molmo-7B (Deitke et al., 2024), finding that pointing-via-text is accompanied by an internal visual search routine, despite the lack of explicit supervision on internal attentional state. We also characterize the algorithm that supports this search routine, and identify a specific set of attention heads, that we term *search heads*, responsible for implementing this algorithm. Finally, we study the consequences of learning to point in tasks that require binding, such as counting and visual search. In experiments with Qwen2-VL-7B-Instruct (Wang et al., 2024), we find that learning to point both eliminates binding errors and facilitates out-of-distribution generalization, suggesting that learning to point-via-text may be an effective solution to the binding problem for VLMs. To summarize, our work makes the following major contributions:

1. We identify the mechanistic and representational origins of binding errors in VLMs, linking these to attentional interference between compositional representations.

2. We demonstrate that pointing-via-text is accompanied by an internal visual search routine, similar to human serial attention.

3. We characterize the algorithm that carries out this search routine, and identify *search heads* as a key mechanism for implementing this algorithm.

4. We show that learning to point-via-text resolves binding errors and enables out-of-distribution generalization in multi-object tasks such as counting and visual search.

Taken together, these results suggest that pointing-via-text may be an effective way to augment VLMs with a capacity for serial visual attention, thus enabling a human-like capacity to perform visual binding.

## 2    RELATED WORK

### 2.1    PARALLEL AND SERIAL PROCESSING IN HUMAN VISION

A large body of work in psychology and neuroscience has identified two distinct modes of processing, one that is largely parallel, feedforward, automatic, and error-prone (sometimes referred to as 'system 1' processing), and one that is serial, effortful, and more precise (sometimes referred to as 'system 2' processing) (Schneider & Shiffrin, 1977; Sloman, 1996; Miller & Cohen, 2001; Kahneman, 2011). The human visual system is also characterized by this dichotomy. In particular, rapid visual processing is thought to be largely limited to feedforward, parallel computations, and is subject to the binding problem, particularly for displays involving a larger number of objects, whereas serial processing enables precise object-centric binding at the expense of longer processing times (Treisman & Gelade, 1980; Treisman & Schmidt, 1982).

More recent work has proposed a normative theory that explains the existence of this tradeoff, identifying *compositional representations* as the source of the binding problem(Frankland et al., 2021a; Musslick et al., 2020; Nurisso et al., 2025). Specifically, it has been proposed that the use of shared (i.e. compositional) representational resources to represent multiple items in parallel (e.g., multiple objects in a scene) produces interference that ultimately leads to binding errors. Compositionality is thought to be an important part of the human capacity for flexible generalization (Fodor & Pylyshyn, 1988; Lake et al., 2017), suggesting that there is a tradeoff between flexibility and parallel processing capacity (Frankland et al., 2021a).

### 2.2    COMPOSITIONALITY AND CAPACITY LIMITS IN ARTIFICIAL NEURAL NETWORKS

Recent studies have identified several notable failure modes of VLMs, particularly involving multi-object tasks such as counting (Rahmanzadehgervi et al., 2024; Rane et al., 2024; Zhang & Wang, 2024), visual search (Campbell et al., 2024), or relational processing (Conwell & Ullman, 2022; Mitchell et al., 2023; Yiu et al., 2024; Fu et al., 2025), and these failures have been explicitly linked to the binding problem (Campbell et al., 2024). At the same time, VLMs and LLMs have both displayed evidence of some degree of compositionality (Lewis et al., 2022; Lepori et al., 2023; Griffiths et al., 2025). One interpretation of these results is that VLMs are susceptible to the binding problem precisely because of their use of compositional representations, consistent with the recently proposed normative explanation of this problem in human cognition (Frankland et al., 2021a). This also suggests that serial processing may be an effective approach to overcome these limitations in VLMs just as it is in human vision.

### 2.3    EXISTING SERIAL PROCESSING CAPABILITIES IN LLMS AND VLMS

Sequentially extended processing has recently played an important role in expanding the reasoning capabilities of language models. Chain-of-thought (Wei et al., 2022), in which language models perform extended reasoning via a sequence of textual outputs, was originally proposed a prompting technique, enabling promising but somewhat limited improvements on reasoning tasks. More recently, reasoning models have been trained to use chain-of-thought more effectively via reinforcement learning (OpenAI & et al., 2024b; DeepSeek-AI & et al., 2025). This is also part of a broader trend toward performing more extensive compute at inference time (Snell et al., 2024; Yao et al., 2023; Webb et al., 2023). Despite these advances in sequential processing for language models, there has been considerably less attention paid toward sequential processing methods for vision language models, though there have been proposals (Yang et al., 2022; Xu et al., 2024; Chen et al., 2021; Sarch

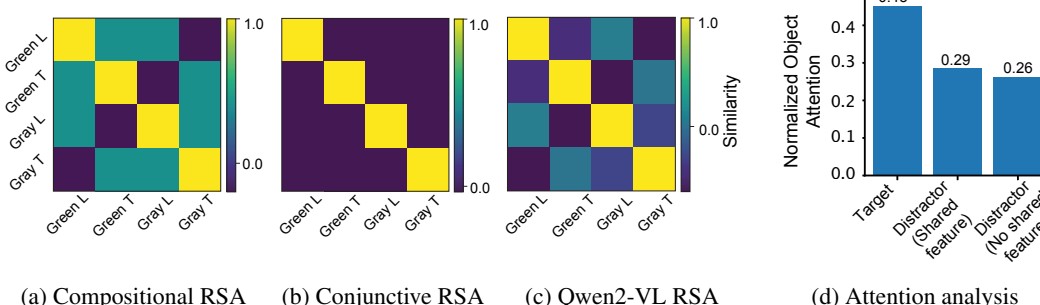

(a) Compositional RSA    (b) Conjunctive RSA    (c) Qwen2-VL RSA    (d) Attention analysis

Figure 2: **Representatoinal and Mechanistic Analysis of Binding Errors in VLMs.** a) Idealized similarity matrix for compositional representations. b) Idealized similarity matrix for conjunctive representations. c) Observed representational similarity matrix for Qwen2-VL-7B-Instruct, displaying higher correlation with compositional model ($r = 0.974$ for compositional model, $r = 0.9$ for conjunctive model). d) Normalized object attention scores reflect the average of the attention scores to each object in one of three categories. Average attention directed toward distractors that share a feature with the target is higher than that directed toward distractors that share no features with the target.

et al., 2025; Cao et al., 2025), including the pointing procedure that we investigate in the present work (Deitke et al., 2024).

## 3 RESULTS

### 3.1 INTERFERENCE FROM COMPOSITIONAL REPRESENTATIONS CAUSES THE BINDING PROBLEM IN VISION LANGUAGE MODELS

To better understand the binding errors displayed by VLMs (Campbell et al., 2024), we investigated the representations and mechanisms employed in a visual search task (see Section A.2.1). In this task, an array of multiple colored shapes is presented, and the task is to identify whether a target object is present (e.g., a green T). An array of distractor objects is also presented, each possessing one of the target features (e.g., a green L or a red T). The binding problem in human vision is thought to arise due to interference between compositionally structured representations. In other words, if the representations of the distractors are partially overlapping with the representations of the target object (i.e., *compositional* codes), this will produce interference when trying to locate the target object. This is in contrast to *conjunctive* representations, in which each conjunction of features is assigned a distinct representation, preventing interference and avoiding the binding problem.

We tested the hypothesis that the binding problem arises in VLMs due to interference between compositional representations, focusing our analysis on the Qwen2-VL-7B-Instruct VLM (Wang et al., 2024). First, we examined the representations used by the model, using representation similarity analysis (RSA) (Kriegeskorte et al., 2008). We extracted visual object embeddings for the targets and distractors in a visual search task, and computed a similarity matrix representing the pattern of pairwise similarities between each feature conjunction (Figure 2c). We also computed idealized similarity matrices representing compositional (Figure 2a) and conjunctive (Figure 2b) representations. The similarity matrix for the model more closely resembled the expected pattern for compositional representations, due to the intermediate level of similarity for pairs of objects that share one but not both features. This was confirmed by a correlation analysis (compositional RSA: $r = 0.974$; conjunctive RSA: $r = 0.9$), indicating the model employs compositional representations.

We then investigated the impact that these representations have on the distribution of attention. Our hypothesis was that binding errors arise due to interference from partially overlapping representations. To test this, we used a modified form of the visual search task in which some distractors are completely distinct from the target object, and other distractors share a single feature. We found that attention was indeed primarily directed to distractors that share a feature with the target object (Figure 2d). These results provide a richer picture of the binding errors that arise for feedforward processing

in VLMs, confirming that these errors arise due to attentional interference between compositional representations.

## 3.2 TRAINING TO POINT INDUCES SERIAL ATTENTION

We next investigated pointing-via-text as a candidate mechanism for serial attention in VLMs. In that approach, a VLM is trained to produce reasoning traces that involve explicit spatial coordinates for the objects in the scene. Although this approach intuitively resembles serial spatial attention, it is important emphasize that there is no direct supervision on the internal attentional states, meaning that there is no guarantee that learning to point actually induces a model to serially attend to objects.

To investigate this, we performed a mechanistic analysis of Molmo-7B, an open-source VLM that has been trained to point. We first computed a summary visualization of the model's attention states while generating a pointing trace. We prompted the model to perform a counting task involving real-world images, and visualized the average attention distribution corresponding to each set of coordinates. The results revealed a highly interpretable pattern, with attention focally directed to one object at a time (Figure 1), consistent with serial visual attention.

To quantify this, we systematically compared the centroid of each attention distribution to the coordinates generated in the model's textual output. Attention centroids were computed on a layer-by-layer basis as follows:

$$x_{\text{cen}} = \frac{\sum_{(x,y)} x \, A(x,y)}{\sum_{(x,y)} A(x,y)} \tag{1}$$

$$y_{\text{cen}} = \frac{\sum_{(x,y)} y \, A(x,y)}{\sum_{(x,y)} A(x,y)} \tag{2}$$

The attention maps $A(x,y)$ are denoised as follows:

$$A(x,y) := \begin{cases} A(x,y), & \text{if } A(x,y) > 0.9 \, M \\ 0, & \text{otherwise} \end{cases} \quad \text{where} \quad M := \max_{(x,y)} A(x,y) \tag{3}$$

The final attention centroids for each point were obtained by averaging over all tokens corresponding to that point:

$$\bar{x}_{\text{cen}}^{(p)} = \frac{1}{|p|} \sum_{t \in p} x_{\text{cen}}^{(t)} \tag{4}$$

$$\bar{y}_{\text{cen}}^{(p)} = \frac{1}{|p|} \sum_{t \in p} y_{\text{cen}}^{(t)} \tag{5}$$

where $(x_{\text{cen}}^{(t)}, y_{\text{cen}}^{(t)})$ is the attention centroid at generation time point $t$, $p$ is the set of tokens corresponding to the generated point $p$, and $|p|$ is the number of tokens in that set. We then computed the root-mean-square error (RSME) between these attention centroids and the corresponding coordinates in the model's textual output.

Figure 3b (gray line) shows the results of this analysis. The correspondence between the model's internal attention and externally generated coordinates was highest in intermediate layers, with RSME reaching the lowest point in layer 19. Earlier and later layers had comparably higher RSME. These results confirm that the model performed an internal serial search while generating textual coordinates, with this visual search most strongly represented in intermediate layers.

### 3.2.1 MECHANISMS OF SERIAL ATTENTION: SEARCH HEADS

To establish a causal link between the model's internal attention states and its externally generated coordinates, we also performed causal mediation analysis (Pearl, 2022; Wang et al., 2022; Meng

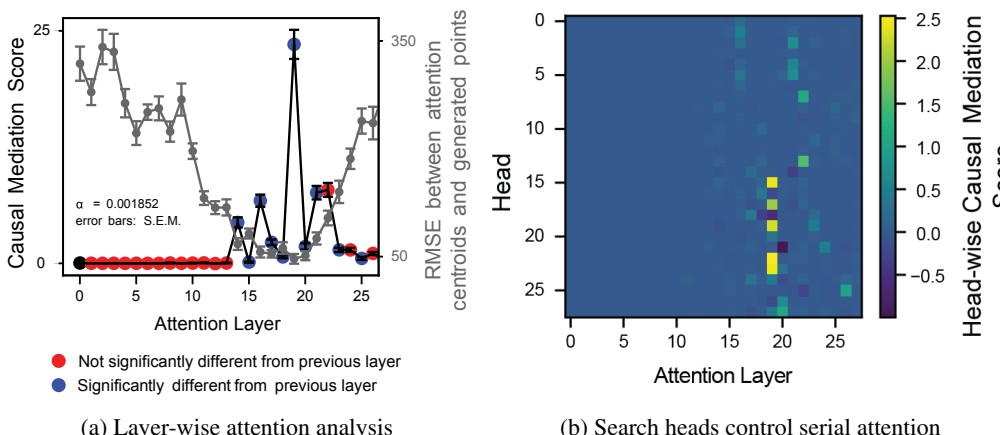

(a) Layer-wise attention analysis        (b) Search heads control serial attention

Figure 3: **Mechanisms of serial visual search in Molmo-7B.** a) We examined (1) the correspondence between the distribution of internal attention and the model's externally generated coordinates (RMSE, gray line), and (2) the causal effect of the attentional state on those coordinates (causal mediation score, black line). Both measures displayed a peak in layer 19. $\alpha = 0.05/27 = 0.00185$ is the Bonferroni-corrected significance level (Weisstein, 2004). b) Head-wise causal mediation scores reveal a set of *search heads* in layer 19 that control the serial attention routine.

et al., 2022; Yang et al., 2025). For each image, we sampled a pointing trace from the model, and randomly selected two points A and B from this trace. We then replaced the attention map for point A with the attention map from point B, and computed a causal mediaton score reflecting the extent to which this intervention made the model more likely to generate the coordinates associated with B. This analysis was performed in a layer-by-layer manner, estimating the extent to which the attentional state in each layer had a causal influence on the model's pointing behavior.

The results of this analysis are shown in Figure 3a (black line). Causal mediation scores showed a sharp peak in layer 19, corresponding precisely to the dip in RMSE indicating a close match between the centroid of attention and the generated coordinates. These results revealed a process centered in layer 20, for which attention both matches the model's generated coordinates, and that is causally responsible for generating those coordinates.

To investigate this process further, we performed causal mediation analyses at the level of individual attention heads (Fig. 3b). The results revealed a set of attention heads in layer 19 with a significant causal impact on the coordinates generated by the model. We refer to these attention heads as *search heads*, as they are causally responsible for controlling the model's serial visual search.

### 3.2.2 ALGORITHMIC-LEVEL INTERPRETABILITY ANALYSIS

We next sought to characterize the algorithm that Molmo uses to perform visual search. We considered three causal mediation scenarios: (1) replacing the attention maps of point A with those of point B (later in the pointing trace), (2) replacing the attention maps of point B with those of point A (earlier in the pointing trace), and (3) interchanging the attention maps corresponding to points A and B (Figure 4). We conducted experiments and reported results for multiple randomly sampled pairs of points.

The results revealed that model possesses an intrinsic order, and maintains a record of objects that have already been counted. When patching from position B (later in the sequence) to position A (Figure 4, left), the model then proceeded to the next object following position B. When patching from position A (earlier in the sequence) to position B (Figure 4, middle), the model did start counting from A again, but instead continued to B followed by the subsequent points. When interchanging A and B (Figure 4, right), the model showed a more complex pattern. After counting B, the model proceeded to the object following B. However, after counting A, the model did proceed to the object following A, but instead continued the count starting from the already-counted object that was farthest

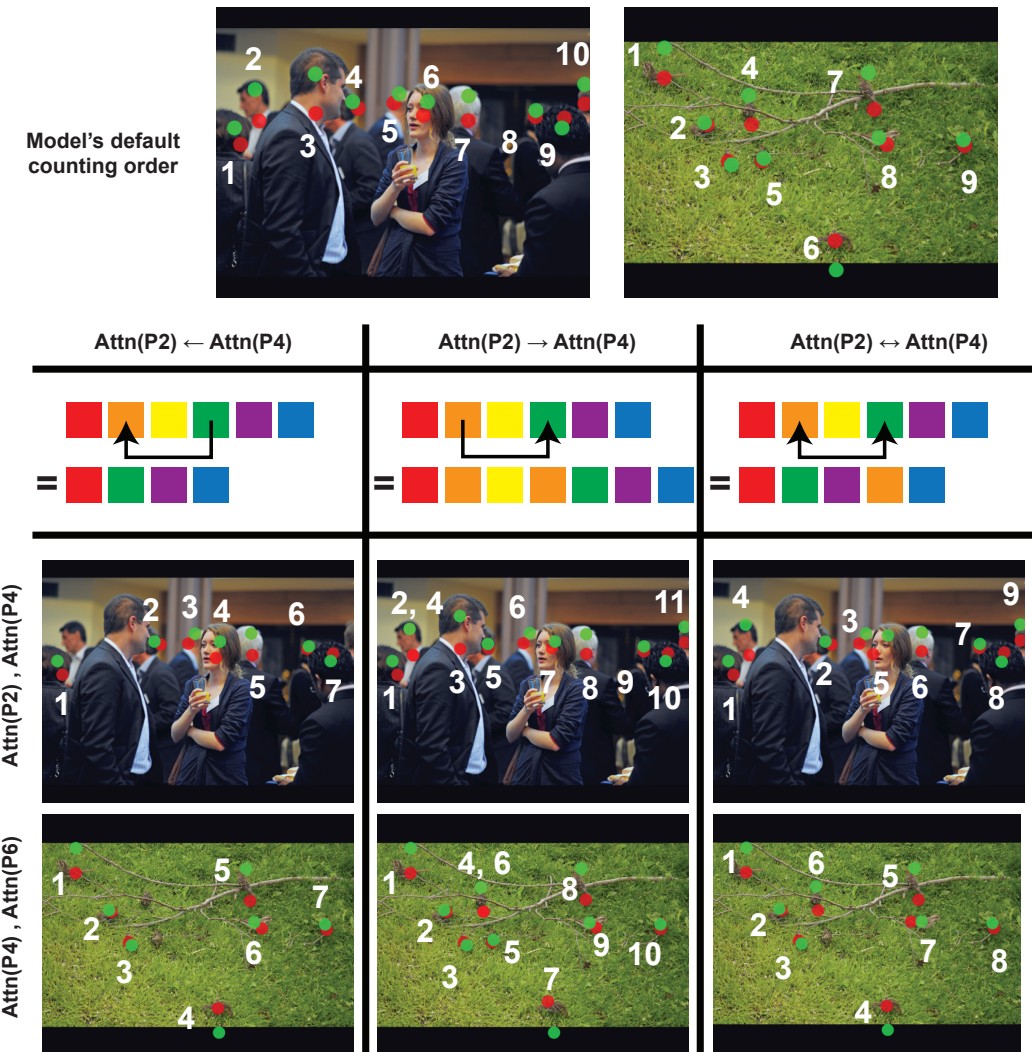

Figure 4: **Characterizing the algorithm of serial visual search.** Annotations in the two images in the first row show the default order in which the model attends to objects. Red and green dots represent the average attention centroid and the generated coordinates in the output text, respectively. We conducted causal mediation analysis in three ways, each illustrated in a separate column. Left: replace the attention map of point A with those of point B (later in the sequence). Middle: replace the attention maps of point B with those of point A (earlier in the sequence). Right: interchange the attention maps of points A and B.

along the intrinsically ordered sequence. These results reveal an intricate and systematic algorithm governing the model's visual search routine.

## 3.3 SERIAL ATTENTION SOLVES THE BINDING PROBLEM FOR VLMS

Finally, we directly investigated whether learning to perform serial attention (by training to point-via-text) can resolve the binding problem for VLMs. We used the Qwen2-VL for this analysis rather than Molmo, as we sought to train a model to perform pointing from scratch (rather than studying a model that had already been trained to point) so that we could control the distribution of training and test data. We fine-tuned models to solve two tasks – counting and visual search – that are known to require binding, and on which VLMs display systematic deficits. We investigated two fine-tuning modes: 1) *direct answer*, in which the model is trained to perform a given task by generating the

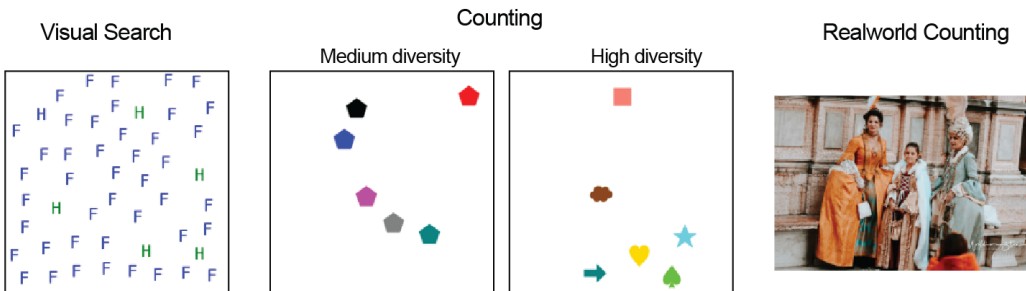

Figure 5: **Examples from the datasets.** The Visual Search task requires the model to identify whether the target (i.e., Blue H) is present in the image. The Counting task requires the model to count the number of objects in the scene (i.e., 6). The Real-world Counting task requires the model to count the number of instances of a given object (i.e., People, 4).

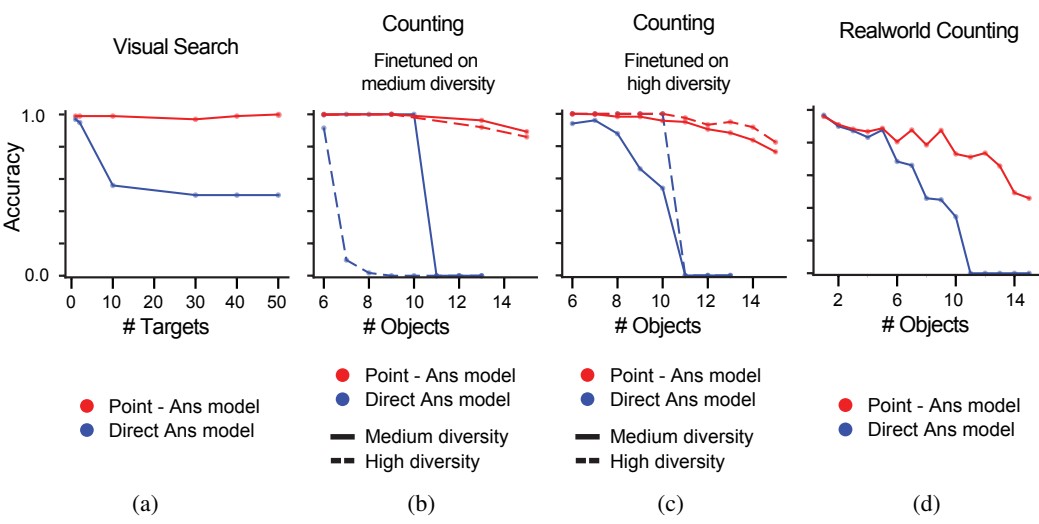

Figure 6: **Serial processing eliminates binding errors and enables OOD generalization.** The point-to-answer model a) generalizes to higher number of targets in a visual search task, b,c) generalizes to counting larger numbers of objects with out-of-distribution image statistics, and d) generalizes to counting larger numbers of objects in real-world images.

answer in a single forward pass, and 2) *point-to-answer*, in which the model is trained to first point to all the objects in an image (stating the corresponding spatial coordinates for each object) and then generate an answer. Example prompt–answer pair for the direct answer model for the visual search task is as follows (see section A.2 for more details).

---

**Point-to-Answer Model (Visual Search)**

**Prompt**: You are presented with an image containing a set of objects, specifically the objects "F" and "T". These objects will appear in either blue or gray. Your task is to determine if there are any gray "F"s in the image. Follow these steps carefully: 1. Please point to each object one at a time, describing the color and shape of each object after you point to it. 2. If the F appears in gray, immediately conclude your response by stating [True]. Otherwise, conclude the response by stating [False]. Enclose your final answer in square brackets.

**Answer format**: x1="4.3", y1="16.4", "shape": "T", "color": "gray", x2="4.3", y2="30.1", "shape": "F", "color": "gray". [True]

---

We made the two following predictions: 1) Both models should learn to perform the task with high accuracy, but 2) only the point-to-answer model should generalize out-of-distribution. This is because we anticipated that the direct answer model would be able to solve the task by exploiting statistical regularities in the training data, thus avoiding the binding problem, but also preventing it from learning a generalizable solution. In contrast, we predicted that the point-to-answer model would be able to avoid the binding problem via serial attention while retaining the benefits of generalizable, compositional representations, thus enabling it to generalize to out-of-distribution task configurations.

Figure 6a shows the results for the visual search task (Figure 5: visual search). Models were fine-tuned on problems involving only a single visual target, and tested on problems involving larger numbers of objects. Both models achieved nearly perfect accuracy for problems with a single target, but only the point-to-answer model generalized to problems involving more targets. This is especially striking given that increasing the number of targets should make the task easier, suggesting that the direct answer model did not genuinely learn to perform visual search, but instead learned to exploit the statistical regularities of the task.

Figure 6b and 6c show the results for the counting task (Figure 5: counting). In this task, we manipulated the diversity of the objects presented within an image. In one setting, models were trained on images with medium diversity and tested on images with high diversity. In another setting, this order was reversed. Additionally, all models were trained on images involving 5-10 objects, and tested on images involving 11-15 objects. The point-to-answer model generalized well in all conditions, generalizing both to different levels of diversity and larger object counts, whereas the direct answer model did not generalize well in any of these conditions, with performance dropping to 0% for larger object counts. These results again suggest that the direct answer model did not actually learn to count, but instead exploited statistical regularities in the task.

Finally, we also trained models to perform counting with real-world images (Figure 6d, Figure 5: real-world counting), finding similar albeit less pronounced effects – the point-to-answer model again showed stronger generalization to larger counts (although both models struggled even with in-distribution counts, presumably reflecting the greater difficulty of this task).

Taken together, these results confirm our core predictions: learning to serially attend to objects one at a time (by pointing to them) enabled the model to evade the binding problem while maintaining the generalizable benefits of its compositional representations, whereas learning to perform the task in a single forward pass was only possible via statistical shortcuts that did not generalize out-of-distribution. These results suggest that pointing-via-text may be an effective solution to the binding problem for VLMs.

## 4    CONCLUSIONS AND FUTURE WORK

In this work, we have presented a comprehensive investigation of pointing-via-text as a potential solution to the binding problem faced by current VLMs. Our results indicate that learning to point induces an internal visual search routine, supported by a newly identified set of attention heads – search heads – that implement a sophisticated algorithm for keeping track of objects. We also demonstrate that learning to point enables VLMs to both eliminate binding errors and generalize out-of-distribution, thus evading the binding problem while maintaining the benefits of compositional representations.

These results constitute a proof-of-principle that training VLMs to point may be an effective solution to the binding problem, but a number of open questions remain concerning how to apply this training procedure in a general-purpose manner. The training procedure investigated in the present work involves supervised fine-tuning on procedurally generated pointing traces. This approach works well for specific tasks, but is unlikely to be feasible for a broader distribution of tasks. This is in contrast to recent training techniques for reasoning models, that avoid the need for supervised reasoning traces by training models using reinforcement learning (DeepSeek-AI & et al., 2025). An important question for future work is how serial visual attention might be incorporated into more general-purpose training procedures for visual reasoning.

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

# A APPENDIX

## A.1 MODELS

For the intepretability analysis, we used Molmo 7B-D (8.02B params). For the finetuning experiments, we used Qwen2-VL-7B-Instruct (8.29B params). Finetuning is conducted on full-weight model without low-rank adaptations. Optimizer is AdamW with learning rate of 1e-5, cosine annealing learning rate scheduler with 200 warm-up steps.

## A.2 DATASETS

### A.2.1 VISUAL SEARCH

The visual search dataset contains scenes with 50 objects, where positive scenes have 1 target and 49 distractors, while negative scenes have 50 distractors.

The colors and shapes of the objects are sampled from $\{red, green, blue, purple, gray, black\}$ and $\{L, T, H, E, F, \Gamma\}$, respectively.

For finetuning, we randomly select a target conjunction for each scene from $\{(red, L), (green, T), (blue, H), (purple, E), (gray, F), (black, \Gamma)\}$. For testing, we sample scenes with target objects from the unseen test conjunction set $\{(gray, L), (green, E), (green, L), (red, E), (red, F), (gray, T), (blue, F), (black, H), (blue, E), (red, H)\}$.

After selecting a target object for each scene, we populate it with 49 distractors, some of which share the same color as the target and others the same shape. The number of distractors in a scene that share the target's color is sampled from a truncated normal distribution with a mean of 25 and a standard deviation of 12.

We supervised fine-tuned Qwen2-VL on 2000 scenes (e.g., Figure 5: Visual Search) using the corresponding prompt–answer pairs (e.g., A.2.1 and A.2.1)

For OOD generalization tests, we created multiple test sets where the target comes from the test conjunction set, with different numbers of target instances present in the scene (i.e., $\{1, 2, 10, 30, 40, 50\}$). For each condition, we generated 50 scenes.

---

**Point-to-Answer Model (Visual Search)**

**Prompt**: You are presented with an image containing a set of objects, specifically the objects "H" and "F". These objects will appear in either green or blue. Your task is to determine if there are any blue "H"s in the image. Follow these steps carefully: 1. Please point to each object one at a time, describing the color and shape of each object after you point to it. 2. If the H appears in blue, immediately conclude your response by stating [True]. Otherwise, conclude the response by stating [False]. Enclose your final answer in square brackets.

**Answer format**: x1="4.3", y1="83.2", "shape": "F", "color": "blue", x2="5.1", y2="27.7", "shape": "F", "color": "blue", x3="8.6", y3="46.9", "shape": "F", "color": "blue", x4="8.6", y4="93.4", "shape": "F", "color": "blue", x5="9.0", y5="70.7", "shape": "F", "color": "blue", x6="9.4", y6="58.6", "shape": "F", "color": "blue", x7="16.4", y7="20.3", "shape": "H", "color": "blue". [True]

---

**Direct Answer Model (Visual Search)**

**Prompt**: blue H

**Answer format**:[True]

---

### A.2.2 COUNTING

The counting dataset contains scenes with varying numbers of objects. The object colors and shapes are randomly selected from {'red', 'magenta', 'salmon', 'green', 'lime', 'olive', 'blue', 'teal', 'gold', 'purple', 'saddlebrown', 'gray', 'black', 'cyan', 'darkorange'} and {'airplane', 'triangle', 'cloud', 'X-shape', 'umbrella', 'pentagon', 'heart', 'star', 'circle', 'square', 'spade', 'scissors', 'infinity', 'check mark', 'right-arrow'}.

The finetuning set contains scenes with {6, 7, 8, 9, 10} objects, with 400 scenes per condition. Multiple OOD generalization test sets are constructed with {11, 12, 13, 14, 15} objects, with 400 scenes per condition.

An example prompt–answer pair for supervised finetuning for the scene in Figure 5 (High-diversity counting) is shown in prompts A.2.2 and A.2.2.

---

**Point-to-Answer Model (Counting Task)**

**Prompt**: You are presented with an image containing several objects. Your task is to accurately count the number of objects in the image. Follow these instructions carefully: 1. Please point to each object one at a time, describing the color and shape of each object after you point to it. 2. After describing all the objects, conclude your response by providing the total count of objects as an integer, enclosed in square brackets.

**Answer format**: x1="38.3", y1="90.6", "shape": "right-arrow", "color": "teal", x2="41.0", y2="55.5", "shape": "cloud", "color": "saddlebrown", x3="50.4", y3="11.3", "shape": "square", "color": "salmon", x4="58.2", y4="82.8", "shape": "heart", "color": "gold", x5="72.7", y5="92.2", "shape": "spade", "color": "lime", x6="78.5", y6="73.8", "shape": "star", "color": "cyan". [6]

---

**Direct Answer Model (Counting Task)**

**Prompt**: [No prompt]

**Answer format**: [6]

---

### A.2.3 REAL-WORLD COUNTING

We utilized the Pixmo-counting Deitke et al. (2024) dataset to construct a real-world counting dataset. Specifically, we filtered out scenes with number of objects {1, ..., 10} and created the finetuning set with 21,699 examples. OOD generalization test datasets are created by filtering out scenes with number of objects {11, 12, 13, 14, 15}.

Example prompt-answer pairs for the scene in Figure 5 (Real-world counting) are shown in prompts A.2.3 and A.2.3.

---

**Point-to-Answer Model (Real-world Counting Task)**

**Prompt**: You are presented with an image containing people. Your task is to accurately count the number of people in the image. Follow these instructions carefully: 1. Please point to each people one at a time. 2. After pointing to all the people, conclude your response by providing the total count of people as an integer, enclosed in square brackets.

**Answer format**: 1:(30.6, 51.3), 2:(49.2, 63.1), 3:(68.2, 89.0), 4:(69.8, 54.7). [4]

---

---

**Direct Answer Model (Real-world Counting Task)**

**Prompt**: people

**Answer format**: [4]

---

### A.3 METHODS

#### A.3.1 NORMALIZED OBJECT ATTENTION

To obtain Fig. 2d, we generated approximately 7,200 images across 24 random color–shape combinations. Each image contains 50 objects: 1 target, 16–17 distractors sharing the target's color, 16–17 distractors sharing the target's shape, and 16–17 distractors sharing neither target feature. For each image, we computed the average attention over each object category and then averaged these values across images.

#### A.3.2 CAUSAL MEDIATION ANALYSIS

To identify the causality between the attention and the generated points, we perform the causal mediation analysis. We interchange the attention maps corresponding to two generated points and observe whether the output also interchanges. If the generated points are bound to the objects in the scene, the expected outcome should reflect the interchanged generated points.

we consider a modified version of the causal mediation score introduced by Yang et al. (2025) (Eq. 6).

$$s_x = (f_{c^*}(x)[y_{c^*}] - f_{c^*}(x)[y_c]) + (f_c(x)[y_c] - f_c(x)[y_{c^*}]) \tag{6}$$

$f_c(x) \in \mathbb{R}^N$ is the model output for the context $c$ given input $x$. The score was developed to measure how much the logit value $f_c(x)[y_c]$ changes when we interchange the context $c$ with $c^*$. Here, $y_c = \arg\max_y f_c(x)$ is the predicted output for context $c$, and $y_{c^*}$ is the expected output for context $c^*$. $\delta_1 = f_c(x)[y_c] - f_c(x)[y_{c^*}]$ tells us how likely the model is to produce the correct answer $y_c$ given the original context $c$, compared to producing the answer $y_{c^*}$ of a different context $c^*$. $\delta_2 = f_{c^*}(x)[y_{c^*}] - f_{c^*}(x)[y_c]$ indicates how likely the model with the modified context $c^*$ (i.e. $f_{c^*}$) is to produce the expected answer for that context ($y_{c^*}$), in contrast to the original answer $y_c$. The overall impact of changing the context from $c$ to $c^*$ is given by $s = \delta_1 + \delta_2$. A higher causal mediation score indicates that the attention map strongly causes the generated points in the output text.

Let us consider the case in Figure 4, where the attention map of point B is replaced with that of point A. $f_c$ denotes the model at generation time point B without attention modification, while $f_{c^*}$ denotes the model at generation time point B with the attention map replaced by that of point A.

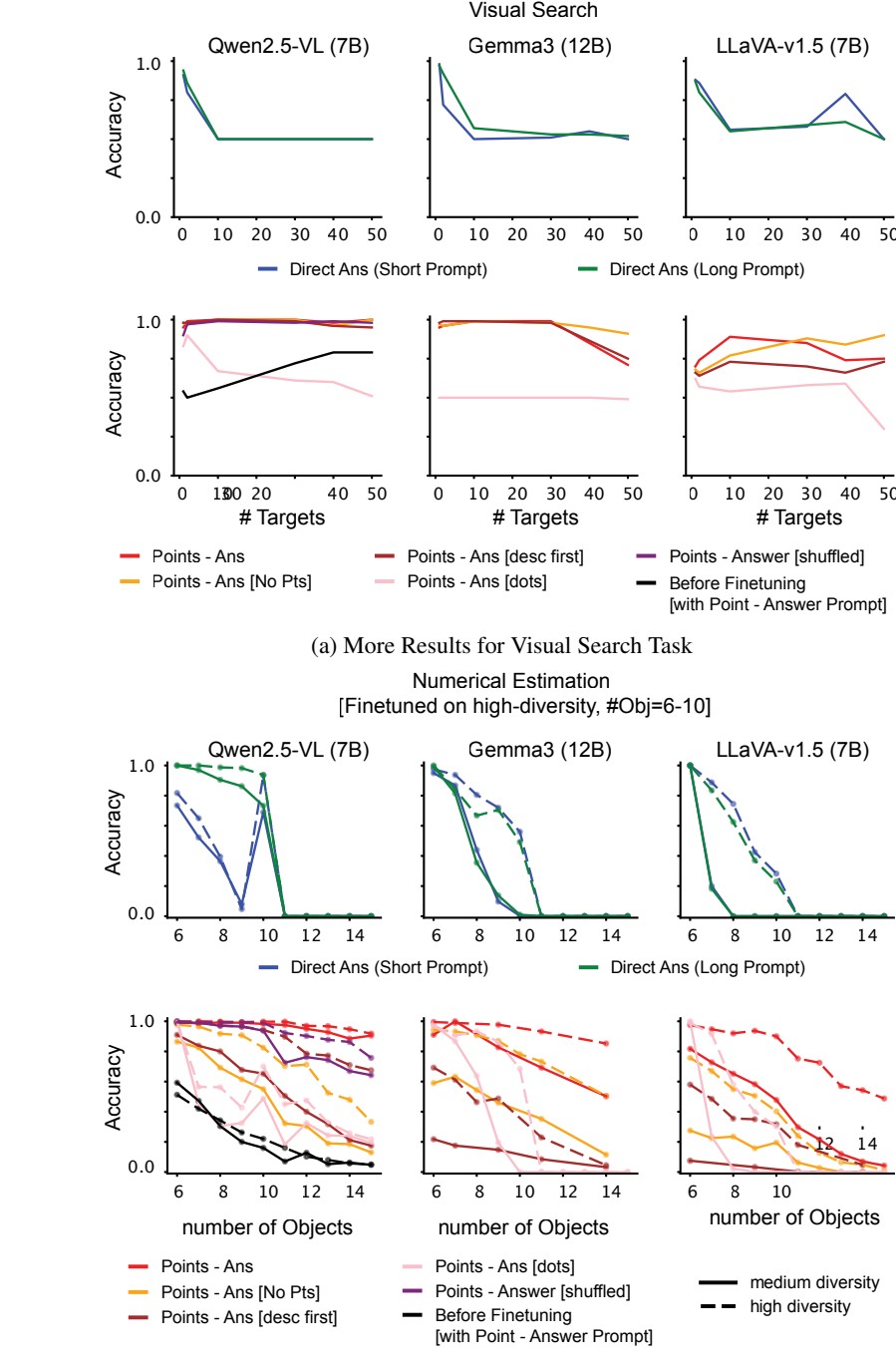

(a) More Results for Visual Search Task

(b) More Results for Numerical Estimation Task

Figure 7: **Point-Ans model outperforms Direct Answer and sequential finetuning baselines.** **Direct Ans (Short Prompt)** uses short prompts and generates short answers. **Direct Ans (Long Prompt)** uses long prompts and generates short answers. **Point-Ans** generates coordinates for each object, followed by their descriptions, and then the final answer. **Point-Ans [No Pts]** generates dots instead of coordinates for each object, along with their descriptions, before producing the final answer. **Point-Ans [desc first]** generates each object's description first, followed by its coordinates, and then the final answer. **Point-Ans [dots]** generates a series of dots followed by the answer. All of the above variants process objects in a top-down, left-to-right order. **Point-Ans [shuffled]** generates points, descriptions, and answers in the same style as Point-Ans, but the object order is randomized. **Before Finetuning** refers to the original model evaluated using the same prompt as the Point-Ans model. See Section A.4.6 for example prompts/ answer formats.

### A.4 MORE ANALYSIS

#### A.4.1 STRONG SERIAL ATTENTION HEAVILY COMES FROM TRAINING-VIA-POINTING WITH COORDINATES

We conducted experiments with more direct-answer baselines and several alternative versions of pointing-via-text models across different classes of models (i.e., Qwen2.5-VL-7B-Instruct, Gemma-3-12B, LLaVA-1.5-7B-HF).

The results for the new direct-answer baseline, *Direct Answer (Long Prompt)*, which uses longer prompts (Fig. 7, green), show that the low performance of direct-answer models is not due to prompt length.

To determine which aspects of Point-Answer finetuning are important, we conducted a series of alternative sequential finetuning methods (Fig. 7). The results indicate that placing a specific coordinate immediately before the object description enables better binding, resulting in the highest performance.

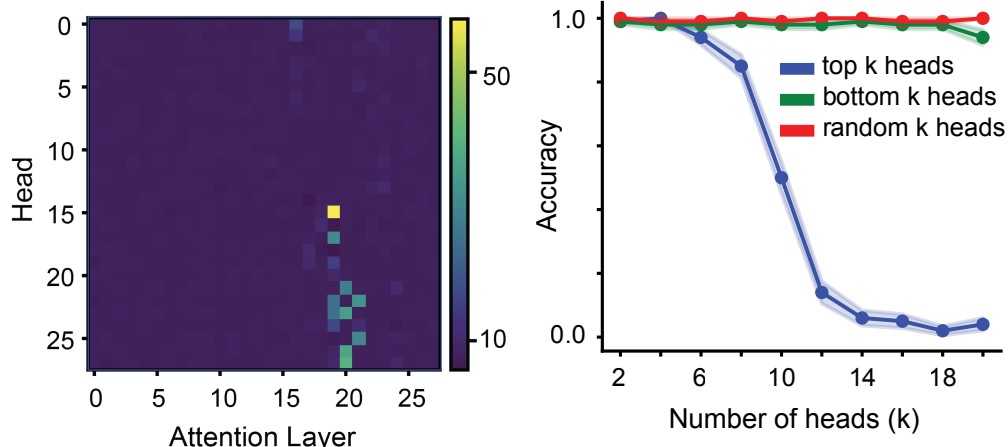

(a) Identified Search Heads on Qwen2.5-VL    (b) Search Heads are Crucial for the Computation

Figure 8: **Search heads emerge in Qwen2.5-VL**. a) To identify search heads, we performed head-wise causal mediation analysis with 37 attention map replacements (across 13 images, with ≈ 3 random point pairs per image). b) We masked out the top-k identified search heads and measured the accuracy over 100 examples. We also conducted the same analysis using the bottom-k and random-k search heads as controls.

#### A.4.2 SEARCH HEADS EMERGE ACROSS DIFFERENT CLASSES OF MODELS, ENABLING SIMILAR PERFORMANCE TRENDS VIA TRAINING-TO-POINT

We repeated the analysis presented in Section 3.2.1 on Qwen2.5-VL and found that search heads also emerge in this model (Fig. 8a).

We further showed that similar performance trends appear across different classes of models, as illustrated in Fig. 7.

#### A.4.3 ABLATING SEARCH HEADS HURTS PERFORMANCE

To evaluate the importance of the search heads identified in Section A.4.2, we ablated the top k search heads. As shown in Fig. 8b, these heads are critical for the task. Ablating even a few of them leads to a substantial drop in performance.

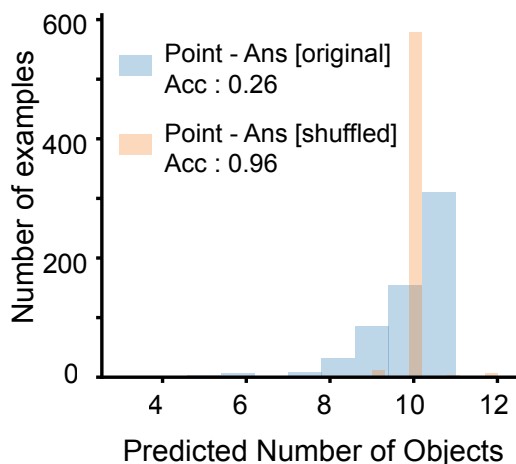

Figure 9: **Point-Ans [Shuffled] model is robust to causal mediation**. We performed causal mediation analysis on both the Point-Ans and Shuffled Point-Ans models using 200 examples from the Counting Task with 10 objects. For each example, we randomly selected three point pairs for the attention replacements, resulting in a total of 600 attention replacements.

### A.4.4 ROBUST COUNTING VIA SHUFFLED POINTING

To examine the effect of shuffling the object order in the Point-Ans model, we finetuned Qwen2.5-VL on Point-Ans prompts and answers with the object order randomly shuffled. Although this model did not surpass the performance of the original Point-Ans model (Fig. 7 - purple), it was robust to causal mediation (Fig. 9).

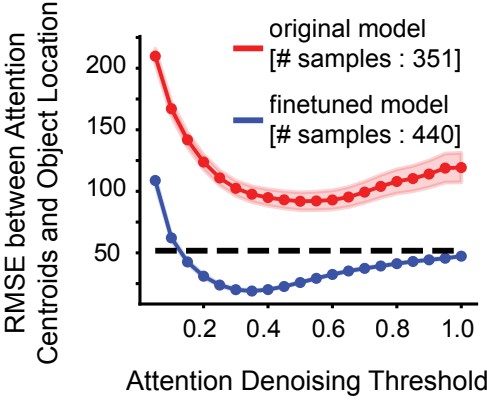

Figure 10: **Point-Answer finetuning allows higher alignment between attention and generated texts**. The attention denoising threshold $\gamma$ is used to denoise the attention maps ($A$) via $A[A \leq A.\max() * \gamma] = 0$. The attention centroid computed over the resulting attention map is then compared with the object location when a valid object is generated in the text. The dotted line represents object size $/2$.

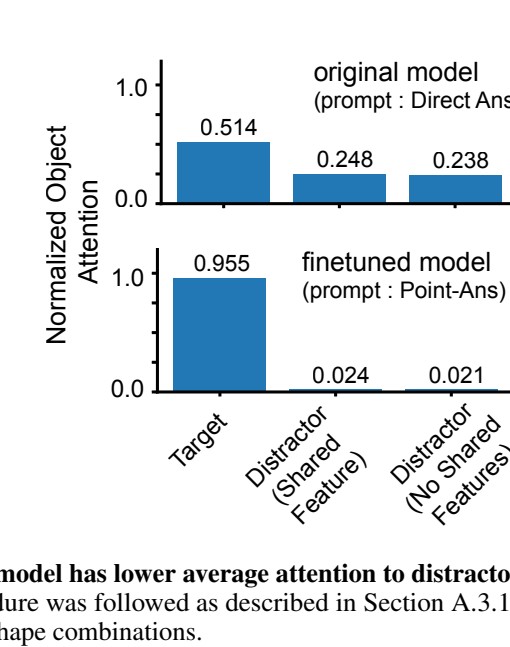

Figure 11: **Point-Ans model has lower average attention to distractors compared to the original model**. Similar procedure was followed as described in Section A.3.1, using 50–60 images across 10–12 random color–shape combinations.

### A.4.5 POINTING FINETUNING IMPROVES TARGET-OBJECT ATTENTION WHILE REDUCING ATTENTION TO DISTRACTORS

To demonstrate that serial attention from point-to-answer finetuning aligns more closely with the generated text, we compared the original base model and the finetuned model under the same point-to-answer prompt. Specifically, we measured the RMSE between the attention centroid and the ground-truth object location for both models (Fig. 10).

Although we saw serial attention in the base model (weaker than the Point-Ans model), its performance on counting-via-pointing is extremely low (Fig. 7 – Black). Therefore, we selected only instances where the model generated the correct objects in the output text, and used the attention maps from those instances to obtain attention centroids and compute the RMSE with the corresponding object locations. We show that point-to-answer finetuning makes the serial attention stronger and better aligned with the output text.

We also show that point-to-answer finetuning suppresses attention to distractors, by computing the average attention over distractors and the target for both the base model and the finetuned model (Fig. 11).

### A.4.6 SUPPLEMENTARY PROMPTS AND ANSWER FORMATS (COUNTING TASK)

---

**Direct Ans (Short Prompt)**

**Prompt**: nan

**Answer format**: [6]

---

**Direct Ans (Long Prompt)**

**Prompt**: You are presented with an image containing several objects. Your task is to accurately count the number of objects in the image. Please respond only by stating the total number of objects. Enclose your final answer in square brackets. Do not provide any additional text or explanation.

**Answer format**: [6]

---

---

### Points-Ans

**Prompt**: You are presented with an image containing several objects. Your task is to accurately count the number of objects in the image. Follow these instructions carefully: 1. Please point to each object one at a time, describing the color and shape of each object after you point to it. 2. After describing all the objects, conclude your response by providing the total count of objects as an integer, enclosed in square brackets.

**Answer format**: x1="29.7", y1="54.7", "shape": "cloud", "color": "cyan", x2="52.0", y2="17.2", "shape": "star", "color": "gray", x3="62.5", y3="11.7", "shape": "airplane", "color": "darkorange", x4="66.0", y4="82.0", "shape": "umbrella", "color": "saddlebrown", x5="66.4", y5="61.3", "shape": "spade", "color": "blue", x6="72.7", y6="44.5", "shape": "check mark", "color": "black". [6]

---

### Points-Ans (No Pts)

**Prompt**: You are presented with an image containing several objects. Your task is to accurately count the number of objects in the image. Follow these instructions carefully: 1. Please point to each object one at a time, describing the color and shape of each object after you point to it. 2. After describing all the objects, conclude your response by providing the total count of objects as an integer, enclosed in square brackets.

**Answer format**: x1=".", y1=".", "shape": "cloud", "color": "cyan", x2=".", y2=".", "shape": "star", "color": "gray", x3=".", y3=".", "shape": "airplane", "color": "darkorange", x4=".", y4=".", "shape": "umbrella", "color": "saddlebrown", x5=".", y5=".", "shape": "spade", "color": "blue", x6=".", y6=".", "shape": "check mark", "color": "black". [6]

---

### Points-Ans (desc first)

**Prompt**: You are presented with an image containing several objects. Your task is to accurately count the number of objects in the image. Follow these instructions carefully: 1. Please describe the color and shape of each object one at a time, and point to each one after you describe it. 2. After describing all the objects, conclude your response by providing the total count of objects as an integer, enclosed in square brackets.

**Answer format**: "shape": "cloud", "color": "cyan", x1="29.7", y1="54.7", "shape": "star", "color": "gray", x2="52.0", y2="17.2", "shape": "airplane", "color": "darkorange", x3="62.5", y3="11.7", "shape": "umbrella", "color": "saddlebrown", x4="66.0", y4="82.0", "shape": "spade", "color": "blue", x5="66.4", y5="61.3", "shape": "check mark", "color": "black", x6="72.7", y6="44.5". [6]

---

### Points-Ans (dots)

**Prompt**: You are presented with an image containing several objects. Your task is to accurately count the number of objects in the image. Follow these instructions carefully: 1. Please point to each object one at a time, describing the color and shape of each object after you point to it. 2. After describing all the objects, conclude your response by providing the total count of objects as an integer, enclosed in square brackets.

**Answer format**: ................................................................................................................................
................................................................................................................................
................................................................................................[6]

## Points-Ans (shuffled)

**Prompt**: You are presented with an image containing several objects. Your task is to accurately count the number of objects in the image. Follow these instructions carefully: 1. Please point to each object one at a time, describing the color and shape of each object after you point to it. 2. After describing all the objects, conclude your response by providing the total count of objects as an integer, enclosed in square brackets.

**Answer format**: x1="62.5", y1="11.7", "shape": "airplane", "color": "darkorange", x2="72.7", y2="44.5", "shape": "check mark", "color": "black", x3="52.0", y3="17.2", "shape": "star", "color": "gray", x4="66.0", y4="82.0", "shape": "umbrella", "color": "saddlebrown", x5="29.7", y5="54.7", "shape": "cloud", "color": "cyan", x6="66.4", y6="61.3", "shape": "spade", "color": "blue". [6]

## Before Finetuning

**Prompt**: You are presented with an image containing several objects. Your task is to accurately count the number of objects in the image. Follow these instructions carefully: 1. Please point to each object one at a time, describing the color and shape of each object after you point to it. 2. After describing all the objects, conclude your response by providing the total count of objects as an integer, enclosed in square brackets.

