# OpenReview forum: "Binding Visual Features Point by Point"
_ICLR.cc/2026/Conference — Submitted to ICLR 2026_

### Official Review · Reviewer_LcCk · 2025-10-27

**Soundness:** 2
**Presentation:** 3
**Contribution:** 2
**Rating:** 4
**Confidence:** 2

**Summary:**

The paper tackles binding errors in vision language models by asking the model to point to each object with xy coordinates before giving the answer. This induces a step by step visual search that reduces mix ups between features and objects. Mechanism work is done on Molmo 7B and shows a tight match between internal attention centroids and the emitted coordinates with a peak effect in middle layers and a small set of search heads that appear to drive the routine. Overall, this paper is a timely and interesting study with a clear and simple pointing strategy and convincing mechanism signals.

**Strengths:**

1. The motivation of the paper is clear and interesting. The research topic is generally interesting and timely.
2. The finding on pointing-via-text is accompanied by an internal visual search routine, similar to human serial attention is a interesting finding.

**Weaknesses:**

1. Empirical experiment coverage is a bit narrow. The mechanism analysis is on Molmo 7B and the training tests are on Qwen2 VL 7B. It make the conclusion and empirical results' generalizability cross family and model scale remain unclear
2. The scope of data is also a bit limited. Synthetic counting uses simple colored shapes. Real world counting focuses on people from Pixmo. something like occlusion and more in the wild counting tasks can strengthen the experiment.

**Questions:**

1. While learning to perform point-via-text can improve binding error and some related tasks, does this affect more general reasoning capacity of VLMs?

Also please refer to weakness section.

---

> ### Author Response · Authors · 2025-11-21
>
> # Testing additional models (Weakness 1)
>
> Thank you for this suggestion. We have now added mechanistic analyses with Qwen2.5-VL (Figure 8a), and training tests with Gemma3 and Llava 1.5 (Figure 7). The mechanistic analyses revealed that training Qwen2.5-VL to point resulted in a similar internal serial search routine as seen in Molmo, and that serial attention was more focused in the fine-tuned model than the base model (Fig. 10, 11). The training tests confirmed that learning to point yielded comparable benefits for the other models tested, indicating that the improvements are not limited to a specific model family.
>
> # Real-world counting (Weakness 2)
>
> The real-world counting experiments performed with the Pixmo dataset do involve real-world images with a significant amount of occlusion and clutter. As an example, for the problem involving counting people shown in Figures 1 and 3, the majority of the people in the image are occluded and appear at a range of depths, many out of focus. This is generally representative of the level of noise and complexity present in these counting problems.
>
> # Generalizability of pointing from supervised fine-tuning (Question 1)
>
> We find that the pointing procedure learned via supervised fine-tuning is somewhat limited to the specific tasks on which fine-tuning data is presented. For instance, Molmo, which has been trained to use pointing for counting, cannot reliably generalize this capability to a visual search task. We suspect that scaling this capability to a more domain-general setting will require other approaches in addition to supervised fine-tuning, such as reinforcement learning with verifiable rewards. One possible path might be to combine supervised fine-tuning at a small scale with reinforcement-learning at a larger scale, analogous to the training procedure that has shown success for text-based LLMs [1]. This approach may allow models to bootstrap from a pointing procedure that is initially learned in a supervised setting, but is then elaborated and extended via RL. Our results in the present work provide motivation for this research direction by demonstrating that pointing can in principle solve the binding problem for VLMs.
>
> [1] Guo, D., Yang, D., Zhang, H., Song, J., Zhang, R., Xu, R., ... & He, Y. (2025). Deepseek-r1: Incentivizing reasoning capability in llms via reinforcement learning. arXiv preprint arXiv:2501.12948.
>
> # Additional results
>
> We have also performed additional experiments that further extend and strengthen our conclusions, including 1) reversible tests (i.e. masking) to show the importance of search heads (Fig. 8b), 2) robust counting via shuffled pointing (Fig. 9).

---

### Official Review · Reviewer_qpJr · 2025-10-28

**Soundness:** 3
**Presentation:** 2
**Contribution:** 2
**Rating:** 2
**Confidence:** 4

**Summary:**

The paper investigates why vision language models often struggle with understanding scenes that contain multiple objects. The authors propose a “pointing via text” approach in which the model identifies each object by providing its spatial coordinates before giving an answer. Analysis shows that this method creates an internal visual search process, which helps them overcome binding errors and improves their ability to generalize to new visual tasks, showing that serial visual processing can effectively solve the binding problem in vision language models just as it does in human vision.

**Strengths:**

1. This paper provides deep interpretability by discovering specific “search heads” responsible for guiding the model’s serial attention, revealed through layer-wise and head-level causal mediation analysis.
2. This paper systematically examines the binding errors that occur in VLMs when processing multi-object scenes and identifies attentional interference as the key mechanism behind these failures.
3. The experiments show that pointing-trained models outperform direct-answer baselines by a large margin across all tested tasks. On both synthetic and real-world datasets, the models that learn to point eliminate binding errors, count objects more accurately, and generalize well to unseen numbers of objects and novel visual configurations.

**Weaknesses:**

1. The experiments focus mainly on counting and visual search tasks, which are simplified and synthetic compared to real-world multimodal challenges. This narrow scope makes it unclear whether the proposed method would scale effectively to more complex visual reasoning or natural images beyond object-level tasks.
2. The “pointing via text” behavior is learned through explicitly supervised coordinate annotations, which are labor-intensive to obtain. This raises questions about scalability and practicality for broader datasets or real-world applications where such supervision is unavailable.
3. The study primarily contrasts the pointing-based model with a single “direct-answer” baseline. It does not compare against other potential solutions to the binding problem, such as attention modulation, iterative reasoning, or reinforcement-based grounding, which would strengthen the empirical claims.
4. The evaluation relies primarily on controlled datasets and small-scale test environments rather than widely recognized multimodal benchmarks. As a result, it’s uncertain how well the approach generalizes across standard large-scale evaluation settings.
5. Although the paper identifies “search heads” within Molmo-7B, it’s unclear whether similar mechanisms emerge in other architectures or model scales. The interpretability findings may thus be model-specific rather than general to vision-language systems.
6. While the paper draws inspiration from human serial attention, it does not quantitatively evaluate whether the model’s attention sequence aligns with human visual search behavior. The analogy to biological vision remains conceptual rather than empirically supported.

**Questions:**

NA

---

> ### Author Response · Authors · 2025-11-21
>
> # Generalization to real-world tasks (Weakness 1)
>
> We would like to note that our experiments include a counting task with real-world images, involving occlusion and clutter, where learning to point provides improvements particularly in the OOD setting (Figure 5d, 6d). This suggests that the method can provide benefits in more complex, real-world settings. Having said this, we agree that generalization to a broader range of visual reasoning tasks is an important priority that will require additional approaches beyond supervised fine-tuning (see next section). Our goal in this work was not to provide a method for scaling this approach, but to evaluate whether pointing can in principle solve the binding problem, which is a major open challenge even for SOTA VLMs [1]. The results suggest that pointing is indeed an effective solution, which motivates the search for more scalable training methods.
>
> [1] Campbell, D. et al. (2024). Understanding the limits of vision language models through the lens of the binding problem. NeurIPS.
>
> # Limitations of supervised fine-tuning (Weakness 2)
>
> We completely agree that supervised fine-tuning has major limitations, and is unlikely on its own to be scalable to support general-purpose visual reasoning. In this work, our goal was to provide a proof-of-principle that learning to point can solve the binding problem for VLMs. However, enabling the use of this capability in more general-purpose, open-ended settings, is a major priority for future research. We envision one possible path for doing so that combines supervised fine-tuning at a small scale with reinforcement-learning at a larger scale, analogous to the training procedure that has shown success for text-based LLMs [2]. This approach may allow models to bootstrap from a pointing procedure that is initially learned in a supervised setting, but is then elaborated and extended via RL. Our results in the present work provide motivation for this research direction by demonstrating that pointing can in principle solve the binding problem for VLMs.
>
> [2] Guo, D. et al. (2025). Deepseek-r1: Incentivizing reasoning capability in llms via reinforcement learning.
>
> We explicitly address these issues in the discussion (Please refer to the Discussion subparagraph starting _The training procedure investigated in the present_ … )
>
>
> # Additional baselines (Weakness 3)
>
> We tested a number of additional baselines, all based on the sequential pointing procedure but with different supervision levels. These include: 1) using the same supervised pointing traces but replacing coordinates with uninformative symbols (“Point - Ans [No Pts]”), 2) swapping the order of the object description and coordinates (“Point - Ans [desc first]”), 3) replacing all tokens in the supervised traces with uninformative tokens (“Point - Ans [dots]”), and 4) shuffling the pointing traces out of raster order (“Point - Ans [shuffled]”). Results are shown in Figure 7 of the appendix. Some baselines improved over the direct answer model, especially the shuffled and No Pts baselines, indicating that strong supervision may not be required for models to develop serial attention. Still, the full pointing model achieved the best overall performance, particularly on counting. Overall, serial attention can emerge without strong supervision, but full pointing traces remain the most reliable way to induce it.
>
> # Evaluation with large-scale benchmarks (Weakness 4)
>
> Given that our goal in this work was to analyze pointing as a potential solution to the binding problem, we chose to focus on a combination of controlled experiments and mechanistic analyses. We have described above some of the challenges that need to be addressed to scale the pointing method to a more task-general setting (as well as some potential solutions), which will likely be necessary to yield more domain-general improvements on large-scale benchmarks. We hope that our results provide a proof-of-principle that will motivate future efforts toward scaling of this approach.
>
> # Additional models (Weakness 5)
>
> In addition to Molmo and Qwen2-VL, we have now also provided results for Qwen 2.5-VL, Gemma3 and Llava1.5 (Figure 7). Learning to point provides substantial improvements for all of the models tested, suggesting that the effectiveness of this approach is not specific to a particular model family.
>
> # Comparison to biological vision (Weakness 6)
>
> Although it would be interesting to evaluate the pointing method as a model of biological vision (e.g. through direct comparison with behavioral or neural data), this was not our goal in the present work. Our goal was to establish whether learning to point solves the binding problem for VLMs. This has a strong conceptual link to biological vision, in that both involve addressing the binding problem via serial visual processing, but we leave the application of this approach as a detailed model of biological vision to future work.

---

### Official Review · Reviewer_VWdX · 2025-11-01

**Soundness:** 3
**Presentation:** 3
**Contribution:** 3
**Rating:** 4
**Confidence:** 3

**Summary:**

This paper tackles the binding problem in VLMs—where features from different objects get mixed in multi-object scenes. It shows that training models to “point via text” (use explicit spatial coordinates) induces an internal visual search routine that sequentially attends to objects, reducing feature swaps and enabling compositional generalization; the pointing behavior then transfers to new tasks with light fine-tuning.

**Strengths:**

1. The visualizations do a great job illustrating the induced “point-via-text” serial search: they make the pre- vs. post-training binding failures immediately visible and show fewer feature–object swaps in multi-object scenes.
2. The paper’s mechanistic analysis is strong: it links pointing supervision to an internal visual search routine and argues that this serial processing reduces binding errors and supports compositional generalization, with behavior transferring to new tasks after light fine-tuning.
3. Performance is strong, with consistent gains over baselines on multi-object benchmarks.

**Weaknesses:**

1. Novelty/positioning. The core idea—training VLMs to output/location-align coordinates—overlaps with prior coordinate- or point-supervised work (e.g., PixelLLM, Pix2Seq, GLIP/Grounding-DINO). The paper should more sharply distinguish its contribution beyond this family [1-5].
2. Supervision & scalability. The method relies on explicit coordinate/point signals (“point-via-text”), which raises annotation-cost and robustness questions for datasets without reliable coordinates; prior literature shows point/coordinate labels are non-trivial to obtain at scale [6].
3. Causal evidence. The paper claims that pointing induces an internal serial search routine, but the support appears mainly correlational; stronger causal interventions/ablations would bolster the claim [7].

[1] Chen et al., Pix2Seq: A Language Modeling Framework for Object Detection (ICLR 2022).

[2] Li et al., GLIP: Grounded Language-Image Pre-training (CVPR 2022).

[3] Liu et al., Grounding DINO (ECCV 2024).

[4] Xu et al., Pixel-Aligned Language Model (PixelLLM) (CVPR 2024).

[5] Yang et al., UniTAB: Unifying Text and Box Outputs for Grounded VL Modeling (ECCV 2022).

[6] Bearman et al., What’s the Point? Semantic Segmentation with Point Supervision (ECCV 2016).

[7] Binding Visual Features Point by Point (OpenReview, 2025).

**Questions:**

Q1: How does the method fundamentally differ from prior coordinate/point-supervised paradigms (Pix2Seq/UniTAB/GLIP/Grounding-DINO/Pixel-LLM) beyond using pointing signals?
Q2: If you disable the discovered “search head(s),” how do binding metrics and task scores change?

---

> ### Author Response · Authors · 2025-11-21
>
> # Clarification on objective of this work (Weakness 1, Question 1)
>
> We would like to clarify that our objective in this work is not to propose a new method, but to analyze the improvements resulting from a previously proposed method (pointing) in relation to the binding problem. Pointing has indeed been proposed in previous work, including the Molmo paper that we explicitly reference. We have also now added references to these other papers in the Related Work. Our contribution is to establish, via mechanistic and representational analyses, that learning to point induces an internal serial attention procedure, and that doing so enables OOD generalization in tasks that are susceptible to the binding problem. This provides a principled perspective on the improvements that result from learning to point and relates these improvements to analogous strategies in human visual processing. Most importantly, these results establish a proof of principle that training to point can solve the binding problem for VLMs (a major limitation even for current SOTA VLMs [1]), and provides motivation for the development of a more general-purpose extension of this training procedure.
>
> [1] Campbell, D., Rane, S., Giallanza, T., De Sabbata, C. N., Ghods, K., Joshi, A., ... & Webb, T. (2024). Understanding the limits of vision language models through the lens of the binding problem. Advances in Neural Information Processing Systems, 37, 113436-113460.
>
> # Limitations of supervised fine-tuning (Weakness 2)
>
> We completely agree that supervised fine-tuning has major limitations, and is unlikely on its own to be scalable to support general-purpose visual reasoning. In this work, our goal was to provide a proof-of-principle that learning to point can solve the binding problem for VLMs. However, enabling the use of this capability in more general-purpose, open-ended settings, is a major priority for future research. We envision one possible path for doing so that combines supervised fine-tuning at a small scale with reinforcement-learning at a larger scale, analogous to the training procedure that has shown success for text-based LLMs [2]. This approach may allow models to bootstrap from a pointing procedure that is initially learned in a supervised setting, but is then elaborated and extended via RL. Our results in the present work provide motivation for this research direction by demonstrating that pointing can in principle solve the binding problem for VLMs.
>
> [2] Guo, D., Yang, D., Zhang, H., Song, J., Zhang, R., Xu, R., ... & He, Y. (2025). Deepseek-r1: Incentivizing reasoning capability in llms via reinforcement learning. arXiv preprint arXiv:2501.12948.
>
> We explicitly address these issues in the discussion:
>
> _The training procedure investigated in the present work involves supervised fine-tuning on procedurally generated pointing traces. This approach works well for specific tasks, but is unlikely to be feasible for a broader distribution of tasks. This is in contrast to recent training techniques for reasoning models, that avoid the need for supervised reasoning traces by training models using reinforcement learning (DeepSeek-AI & et al., 2025). An important question for future work is how serial visual attention might be incorporated into more general-purpose training procedures for visual reasoning._
>
> # Causal interventions (Weakness 3, Question 2)
>
> We would like to clarify that our results do include causal evidence. Specifically, the causal mediation results presented in Figure 3 establish that the internal visual search routine is causally involved in driving the model’s behavior (Note that the review provides a citation (ref 7), but this citation is to our own paper). We have now also performed additional ablation experiments to further bolster the causal evidence. Specifically, we ablated attention heads (by setting their output to 0) and measured the impact on accuracy in the counting task. We ablated either the top-k attention heads (according to the CMA scores), the bottom-k attention heads, or k randomly selected heads, varying k from 2 to 20. The results are included in Figure 8b of the appendix. The results confirmed that ablating the top-k search heads dramatically impaired counting performance, with accuracy falling close to 0% after ablating just the top 14 heads, while performance remained close to 100% for the bottom-k and random-k ablation conditions. This result further demonstrates that the identified search heads are necessary for using visual search to perform counting.

---

### Official Review · Reviewer_eWHG · 2025-11-07

**Soundness:** 3
**Presentation:** 4
**Contribution:** 3
**Rating:** 4
**Confidence:** 3

**Summary:**

The paper focus on binding error in multi-object visual tasks, and examine how the "Point -> answer" protocol improves the tasks both in-domain and ood from the perspective of explanibility. They firstly diagnose binding failures, then connect the performance with the exact layer and "search head", and explore the sequential search mechanism.

**Strengths:**

1. This work focus on a core failure mode for VLMs, binding error, and define it well.
2. Section 3.2.2 is compelling. The test about counting and order shows that the model has a kind of step-by-step search habit, which supports the idea well.
3. The finding locating the behavior at some middle layers and "search heads" gives a real handle for future work on model understanding and reliability.

**Weaknesses:**

1. For section 3.2.1, the interventions are good but if adding reversible tests, like masking out the head or removing its effect to see the performance drop, it would be more reasonable.
2. For section 3.3, the improvements might come from extra training supervision, like using coordinates and multi-step prompts, not from real "serial attention." It would be good to test weaker or noisy training versions.

**Questions:**

1. Section 3.1 argues binding errors comes from compositional-representation–driven attentional interference. Could you re-run the section 3.1 diagnostics on models trained with "Point -> answer" protocol to test whether interference metrics actually improve (e.g., reduced attention bias toward distractors)?
2. Section 3.2.2 is quite interesting. It suggests a left-to-right counting tendency. Is that order imposed by the prompt or most likely aligned with Molmo training data? Can the prompt specify a different direction like right-to-left or top-to-bottom, and does the model follow? In your section3.3 training, which order did you use in your training data? If the training points were shuffled/random-order, what order would "Point -> answer" adopt at test time, and how robust is it to coordinate perturbations like in section 3.2.2 do?

If the authors answer my questions well and make these clarifications, I am happy to raise my review score.

---

> ### Author Response · Authors · 2025-11-21
>
> # Ablation experiments (Weakness 1)
>
> Thank you for this suggestion. We have now added ablation experiments as an additional test of the causal relevance of the identified search heads. Specifically, we ablated attention heads (by setting their state to 0) and measured the impact on accuracy in the counting task. We ablated either the top-k attention heads (according to the CMA scores), the bottom-k attention heads, or k randomly selected heads, varying k from 2 to 20. The results are included in Figure 8b of the appendix. The results confirmed that ablating the top-k search heads dramatically impaired counting performance, with accuracy falling close to 0% after ablating just the top 14 heads, while performance remained close to 100% for the bottom-k and random-k ablation conditions. This result further demonstrates that the identified search heads are necessary for using visual search to perform counting.
>
> # Additional baselines (Weakness 2)
>
> We have now tested a number of additional baselines, all involving variations on the sequential pointing procedure, but with varying degrees of supervision. These baselines include: 1) using the same supervised pointing traces, but with coordinates replaced by uninformative symbols (‘Point - Ans [No Pts]’), 2) swapping the order of the object description and the coordinates, such that the objects are described before their spatial coordinates (‘Point - Ans [desc first]’),  3) replacing all tokens in the supervised traces with uninformative tokens (‘Point - Ans [dots]’), 4) shuffling the order of the pointing traces, such that the objects are no longer listed in raster order (‘Point - Ans [shuffled]’). The results are included in Figure 7 of the appendix. Some of these baselines yielded improvements over the direct answer model, particularly the shuffled and No Pts baselines, suggesting that such strong supervision may not be necessary for the models to learn to perform serial attention. However, the complete pointing model still showed the best overall performance, particularly in the counting task. These results suggest that while serial attention may emerge without such strong supervision, training with full pointing traces is the most effective way to reliably induce serial attention.
>
> # Analysis of attention interference in pointing model (Question 1)
>
> We have now added this analysis to Figure 11 of the appendix. The results confirm that significantly less attention is directed to the distractors in the pointing model than the base model, and more attention is directed to the target object. Note also that we have replaced our attention metric with an average over objects rather than a sum (this does not impact the relevant results, but is a more accurate reflection of the attention to each category of object, not influenced by the number of objects of each type), both for the results in the appendix and the results in the main paper.
>
> # Impact of prompt on count order (Question 2.1)
>
> The model was not explicitly prompted to count the objects in any particular order. However, the training procedure, both for Molmo and for the models that we fine-tuned in the default pointing condition, involved supervised pointing traces with objects in raster order. The basis of the model’s preference for raster order is due to these traces.
>
> # Pointing traces with random object order (Question 2.2)
>
> This is a very interesting suggestion. We have now carried out this experiment, training a model with pointing traces involving objects in random order. Interestingly, as you suggest, this model is indeed more robust to the perturbations tested in section 3.2.2. To illustrate this, we tested the model on counting with scenes involving 10 objects. and performed random coordinate perturbations. The results, shown in Figure 9 of the appendix, confirmed that the model trained on shuffled sequences was more robust, reliably producing the correct answer (‘10’) in most cases, whereas the model trained on raster order traces showed a broader distribution of responses, reflecting enumeration errors induced by the coordinate perturbations.

---

### Author Response · Authors · 2025-12-02
**Summary Comment for the AC**

We would like to thank the reviewers for their time spent reviewing our paper. We would like to provide a summary of the concerns we have addressed as well as the corresponding manuscript edits during the rebuttal period. We believe those additions further strengthened our contribution.

## 1. Strengthened Causal and Mechanistic Evidence
* We added new ablation experiments and reversible masking tests, confirming that the CMA-identified search heads are causally necessary for counting performance [Rev-eWHG-W1, Rev-VWdX-W3/Q2]. We also expanded our attention-interference analysis, showing stronger target-focused attention in pointing-trained models [Rev-eWHG-Q1].

## 2. Added Additional Baselines and Robustness Experiments
* We introduced several weaker-supervision pointing baselines, showing that serial attention can emerge with less supervision while full pointing remains most reliable [Rev-eWHG-W2, Rev-qpJr-W3].
* We also trained models with randomized pointing order, finding improved robustness to coordinate perturbations [Rev-eWHG-Q2.2].

## 3. Broadened Model Coverage
* We added new results for Qwen2.5-VL, Gemma3, and LLaVA-1.5 in addition to Qwen2-VL, demonstrating consistent performance trends and similar serial-search mechanisms/ mechanistic analysis results across model families [Rev-qpJr-W5, Rev-LcCk-W1].

## 4. Generalizability/ Scalability Across Models and Sequential Fine-Tuning Variants
1. We clarified that the original results on the PixmoCounting dataset do involve real-world images with occlusion and clutter [Rev-qpJr-W1, Rev-LcCk-W2].
2. We acknowledged that generalizability remains task-specific under pure supervised fine-tuning (e.g., Molmo fine-tuned for counting does not transfer its pointing routine to visual search), emphasizing the need for additional training approaches beyond supervised traces [Rev-LcCk-Q1]. We also acknowledged the limited scalability of supervised pointing and outlined a path combining small-scale supervision with RL for broader generalization [Rev-VWdX-W2, Rev-qpJr-W2, Rev-LcCk-Q1]. However, the purpose of this work was not to provide a novel training method for learning to point, but to investigate the consequences of learning to point w.r.t the binding problem.
3. We clarified that we did not evaluate large-scale benchmarks because our focus was on controlled settings necessary for mechanistic analysis; scaling pointing to more task-general and benchmark-level settings is left for future work [Rev-qpJr-W4].

## 5. Goal of the Work
* We clarified that our goal is not to propose a new method for training models to point, but to analyze how pointing solves the binding problem via induced serial attention [Rev-VWdX-W1/Q1]. Importantly, this provides a proof-of-principle that pointing can solve some of the major limitations that VLMs currently face (stemming from the binding problem), motivating future research on more scalable training approaches.

## 6. Clarification on the Model’s Raster-Order
* We clarified that the model’s raster-order counting behavior arises from the ordering of the supervised pointing traces rather than from the prompt itself [Rev-eWHG-Q2.1].

## 7. Scope on Biological Vision
* We clarified that, although the approach has a conceptual link to biological vision through serial processing, evaluating it as a model of biological vision is not the goal of this work [Rev-qpJr-W6].

---

### Meta-Review · Area_Chair_ngc2 · 2026-01-05

**Summary:**

The key reviewer concerns are:

* Novelty / positioning: overlap with prior point / coordinate-supervised paradigms; contribution framed more as analysis than a new method [VWdX,qpJr]

* Supervision and scalability: reliance on explicit coordinate traces; annotation cost; limited transfer of pointing routine beyond supervised tasks; lack of large-scale benchmark evaluation [VWdX,qpJr,LcCk]

* Empirical scope: focus on counting and visual search (synthetic and limited real-world), uncertain generalisation to broader multimodal reasoning [qpJr,LcCk]

* Causal evidence: stronger interventions required to show that identified 'search heads' causally drive behavior (e.g. masking, ablations) [eWHG,VWdX]

* Attention interference and mechanism: verify reductions in distractor attention and target-focused serial search [eWHG]

* Order effects and robustness: raster-order bias from traces; robustness to coordinate perturbations; behavior under shuffled pointing order [eWHG]

* Biological vision framing: ensure claims remain conceptual and not overextended as a model of human vision [qpJr]

As outlined below, the author rebuttal added additional analyses (head ablations, interference metrics, shuffled-order training, additional baselines and model families) that address many points. Nonetheless, I hold the opinion that concerns on positioning and scalability / generalisation remain only partially resolved. With an overall negative initial reviewer lean, I propose reject.

**Reviewer Concerns:**

* Mechanistic depth: author provide analyses towards linking pointing supervision to an internal serial visual search routine. Further head-level ablations show performance collapse when ablating top-CMA heads c.f. alternatives. I can consider that this has helped to strengthen causal claims.

* Interference metrics: authors re-ran attention-interference diagnostics showing less attention to distractors and more to targets in the pointing model.

* Order robustness: authors clarified raster-order arises from supervised traces and trained with shuffled pointing sequences, which improved robustness to coordinate perturbations compared to raster-order training.

* Baselines and supervision: authors confirm that additional variants (e.g. desc-first, dots, shuffled) show some gains even with weaker supervision. Full pointing remains most reliable.

* Model coverage: authors add new results (Qwen2.5-VL, Gemma3, LLaVA-1.5), providing some initial evidence of consistent trends across model families.

* Scope / claims calibration: authors clarified they do not claim a biological model and further acknowledge limits of supervised fine-tuning. The former point may require some further manuscript phrasing refinement.

Remaining limitations:

* Novelty / positioning: the core training signal (pointing via text, coords) is established in prior work; the paper’s contribution is primarily diagnostic / mechanistic. Stronger differentiation (e.g., reporting of broader tasks where serial procedures uniquely enable generalisation) would help to further strenghten.

* Scalability and generalisation: pointing routines via supervised learning appear task-specific and do not consistently transfer without additional training; the work does not evaluate broader, large-scale benchmarks, leaving practical impact uncertain.

* Empirical breadth: despite Pixmo results and added models, most evidence centers on counting and visual search under controlled settings; it remains unclear how the induced serial procedure fares on more complex, open-ended multimodal tasks.

**Reviewer Scores:**

* Reviewer eWHG: rebuttal directly addressed requests for reversible tests / ablations, interference metrics, and order robustness. A modest upward adjustment could have been possible.

* Reviewer VWdX: causal head ablations and clearer positioning partially address concerns; scalability remains open. No evidence of movement leaves score likely stable.

* Reviewer qpJr: core reservations on scope, scalability, and broader comparisons persist, despite added baselines and model families. The large required score shift seems unlikely.

* Reviewer LcCk: additional models and clarification of real-world Pixmo settings are responsive, but questions on generality and biological framing may remain. Score likely stable.

Decision recommendation

This submission studies the binding problem in vision language models and shows that supervised point via text can induce a serial search routine. The mechanistic evidence and added ablations are reasonably solid, with broader model coverage and robustness checks. However, key reservations persist: (i) incremental novelty relative to prior coordinate or point supervised work, (ii) uncertain scalability and generality beyond supervised traces, including annotation cost, limited cross task transfer, and no large scale benchmarks, and (iii) narrow empirical scope centered on counting and visual search in controlled settings. I feel that thorough resolution of the remaining concerns would materially strengthen the case for acceptance in a future submission.

---

### Decision · Program_Chairs · 2026-01-26

Reject